# Semi-Embedding Zn-Co_3_O_4_ Derived from Hybrid ZIFs into Wood-Derived Carbon for High-Performance Supercapacitors

**DOI:** 10.3390/molecules27238572

**Published:** 2022-12-05

**Authors:** Wanning Xiong, Jie Ouyang, Xiaoman Wang, Ziheng Hua, Linlin Zhao, Mengyao Li, Yuxin Lu, Wei Yin, Gonggang Liu, Cui Zhou, Yongfeng Luo, Binghui Xu

**Affiliations:** 1Hunan Province Key Laboratory of Materials Surface & Interface Science and Technology, College of Science, Material Science and Engineering School, Central South University of Forestry and Technology, Changsha 410004, China; 2Institute of Materials for Energy and Environment, School of Materials Science and Engineering, Qingdao University, Qingdao 266071, China

**Keywords:** semi-embedding, transition metal oxides, hybrid ZIFs, wood-derived carbon, supercapacitors

## Abstract

Transition metal oxides (TMOs) can provide high theoretical capacitance due to the change of multiple valence states of transition metals. However, their intrinsic drawbacks, including poor electrical conductivity, lower energy density, and huge volume expansion, will result in the pulverization of electrode materials and restricted electrochemical kinetics, thus leading to poor rate capability and rapid capacity fading. Composite electrodes based on transition metal oxides and carbon-based materials are considered to be promising candidates for overcoming these limitations. Herein, we reported a preparation method of hybrid ZIFs derived Zn-doped Co_3_O_4_/carbon (Zn-Co_3_O_4_/C-230) particles semi-embedded in wood-derived carbon skeleton for integrated electrodes. A large specific surface area, excellent conductivity, and electrochemical stability provide a larger electrochemical activity and potential window for the electrode. Prepared Zn-Co_3_O_4_@CW-230 electrode (0.6 mm thick) displays ultrahigh area specific capacitances of 7.83 and 6.46 F cm^−2^ at the current densities of 5 and 30 mA cm^−2^, respectively. Moreover, a symmetric supercapacitor assembled by two identical Zn-Co_3_O_4_@CW-230 electrodes delivers a superior area-specific capacitance of 2.61 F cm^−2^ at the current densities of 5 mA cm^−2^ and great energy densities of 0.36 mWh cm^−2^ (6.0 mWh cm^−3^) at 2.5 mW cm^−2^, while maintaining 97.3% of initial capacitance over 10,000 cycles. It notably outperforms those of most carbon-based metal oxides, endowing the Zn-Co_3_O_4_@CW-230 with extensive prospects for practical application.

## 1. Introduction

A supercapacitor (SC) is considered one of the potential candidates in the field of future generations of energy storage devices due to its high specific capacitance, excellent charge-discharge performance, and stable cycle performance [1,2,3,4]. The innovation of electrode materials is a key link to improving the performance of supercapacitors. In recent years, to further improve the electrochemical properties of supercapacitors, considerable efforts have been devoted to developing new electrode materials with higher specific capacitance and cyclical stability. Recently, some emerging nanocomposite materials (i.e., wood [5], nanocellulose [6], MOFs [7,8], and Mxene [9], etc.) are increasingly used as electrode materials to apply in energy-relative fields.

Among them, biomass-derived carbon-based electrodes have attracted extensive attention and application due to their low cost, rich sources, environmental protection, and superior capacitive energy storage performance [10,11,12,13]. Wood composed of well-connected hollow fibrous structures exhibits abundant oxygen-containing functional groups [14] that can act as active sites for chemical modifications facilitating the in-situ functionalization with polymers or inorganic materials [15,16,17]. Wood-derived carbon inherits the layered honeycomb structure and interconnected large porosity of natural wood and can provide multi-scale directional channels for ion transport [18,19,20,21]. In addition, unlike toner, wood-derived carbon can be directly used as self-supporting electrodes without collectors, additives, and adhesives [22,23]. These show that wood-derived carbon is an ideal 3D framework substrate to support electroactive components as freestanding electrodes for high-performance energy-storage devices.

Transition metal oxides (TMOs) mainly contribute higher capacitance to the whole capacitor due to the change of multiple valence states of transition metals. However, metal oxides have the disadvantages of low electronic conductivity, limited stability, and rapid loss of their capacity during cycling [24,25]. In order to solve these problems, combining TMOs and wood-derived carbon is a surprising way to greatly improve their electrochemical performance. For example, Wang et al. prepared a ZnO-loaded/PC composite [26]. Deng et al. synthesized a Co_3_O_4_/3DGN/NF hybrid [27]. Li et al. prepared Co_3_O_4_ nanospheres embedded in nitrogen-doped carbon framework [28]. They improved the electrochemical properties of composites by surface attachment, anchoring, or fully embedding TMOs into the carbon skeleton.

Zeolite imidazole framework (ZIF) is a well-known MOF structure with high surface area, easy synthesis at room temperature, and inherent nitrogen doping on carbon [24,25,26,27]. In recent years, the combination of ZIFs and wood is considered to be a challenge full of opportunities [29,30]. In this paper, we report a simple and general strategy for preparing hybrid ZIFs derived Zn doped Co_3_O_4_ semi-embedded in wood-derived carbon skeleton (Zn-Co_3_O_4_@CW-230) composite electrode. Compared with ZIF-67, which is easy to collapse under high-temperature calcination, the Co^2+^-excess bimetallic hybrid Co/Zn ZIFs (HZs) [31] is more stable in structure. HZs-derived carbon acts as the electrical highway and mechanical backbone to prevent Co_3_O_4_ from gathering so that the Zn-Co_3_O_4_@CW-230 electrode has excellent electrochemical activity. Compared with surface composites and fully-embedded composites, Zn-Co_3_O_4_/C-230 particles greatly increase the specific surface area of carbonized wood, and semi-embedded structure can improve the stability of electrode materials and shorten the electron/ion transport path, accelerating the reaction kinetics.

## 2. Results and Discussion

Natural wood (poplar wood) can be considered a biopolymer nanocomposite that exhibits well-aligned wood cell walls. The wood cell walls are generally composed of oriented cellulose microfibrils and the matrix of lignin and hemicellulose [32]. As shown in Figure 1, first, the pre-cut poplar slices were pretreated with an alkali solution. This process simultaneously fulfills two main functions: (i) it ensures the ion exchange of the proton in the wood inherent carboxyl groups by sodium cations, providing the nucleation sites for the subsequent growth of the HZs structure, and (ii) it provides a rough fibrillar structure that boosts the anchoring of the HZs to the wood [33]. Then, a layer of HZs was semi-embedded on the inner and outer surfaces of the pretreated wood by the vacuum impregnation method, which is helpful in embedding its derivatives in the carbonized wood in situ. The second step: the Zn-Co_3_O_4_@CW-230 was obtained by consecutive carbonization and oxidation at different calcination temperatures. Semi-embedded Zn-Co_3_O_4_/C-230 particles can be used as energy storage nodes, while wood-derived carbon can be used as a collector and support to promote full contact between electrolyte and electrode materials and rapid transfer of electrons/ions.

The scanning electron microscopy (SEM) images (Appendix A) display a typical natural wood structure for carbonized wood with vertical channels and hierarchical porosity. On the internal section of HZ@Wood, it can be observed that a layer of dodecahedral HZs is evenly semi-embedded on the inner and outer surfaces of the wood (Figure 2a–c). Compared with ZIFs of a single metal base, during the calcination of Co^2+^-based ZIF-67, its polyhedral structure will collapse due to the formation of Co nanoparticles (NPs) (Appendix A) [34,35]. In Appendix A, it can be observed that the wood-derived carbon skeleton of Zn@CW-230 is smooth and complete, and there is no embedded active substance, which proves that ZIF-8 cannot penetrate the wood-derived carbon skeleton. Excitingly, Zn-Co_3_O_4_/C-230 particles can maintain the polyhedral morphology of ZIFs well, although their size shrinks slightly due to the decomposition of organic ligands (Figure 2a–i). Among them, by further enlarging the wood-derived carbon skeleton, it can be clearly seen that the dodecahedral particles are embedded in the wood-derived carbon skeleton, and the SEM energy mapping images show that the Co and Zn elements go deep into the wood-derived carbon skeleton (Figure 2j). The content of active substances in the material was analyzed by TG data. In Appendix A, the weight loss curve of Zn-Co_3_O_4_@CW-230 shows that the content of active substances in the sample is 11.46 wt%. According to the energy-dispersive X-ray spectrogram, the relative atomic ratio (%) of the Zn-Co_3_O_4_@CW-230 sample is obtained (Appendix A). [36] The results show that the oxidation of Co NPs is dominant in the oxidation process, Zn exists in the form of doping, and its atomic percentage is very low (0.21%).

In the XRD pattern (Figure 3a), the main diffraction peaks of Zn-Co@CW come from carbon and Co, which are basically consistent with WC/Co-800 [37]. The Zn-Co@CW and Zn-Co_3_O_4_@CW-230 samples deliver two broad diffraction peaks at ~26° and ~45°, assigned to the (004) and (102) planes of carbon (PDF no. 26-1080). It can be seen from the spectrogram that after carbonization, the crystallinity of wood embedded in HZs is higher than that of pure wood. This is because a large amount of HZs-derived carbon with higher crystallinity will be derived on the electrode surface. In this process, the Co diffraction peak (PDF no. 15-0806) disappears, and the Co_3_O_4_ (PDF no. 43-1003) diffraction peak appears, indicating that Co NPs are indeed oxidized to Co_3_O_4_. Interestingly, Co NPs originated from HZs, which indicates that HZs grow successfully on wood. Figure 3b shows the Raman spectrum of CW, CW-230, Zn-Co@CW, and Zn-Co_3_O_4_@CW-230 samples. The D band around 1340 cm^−1^ corresponds to defective/disordered carbon, while the G band around 1590 cm^−1^ and the 2D band around 2690 cm^−1^ correspond to typical peaks of sp^2^ hybridized graphitic carbons observed in Raman spectra of Zn-Co_3_O_4_@CW-230 [35]. The ratio of D and G peak strengths (I_D_/I_G_) is used to analyze the defects and crystallinity of carbon material. The I_D_/I_G_ ratio of Zn-Co@CW and CW is the same at 0.93. After oxidation in air, the ratio of Zn-Co_3_O_4_@CW-230 and CW-230 decreases because more crystalline carbon is exposed due to the decomposition of amorphous carbon on the electrode surface. It is worth noting that the I_D_/I_G_ ratio of Zn-Co_3_O_4_@CW-230 is reduced to 0.85 may be because the electrode surface with a semi-embedded structure is more conducive to the decomposition of amorphous carbon. This may mean greater conductivity and less resistance.

The chemical composition and valence of all samples are analyzed by the XPS test. No signal peak of Zn can be detected in the survey spectrum because the Zn-relative components are very low (Figure 3c). However, the peaks at 1021.88 and 1045.05 eV can be detected in Zn@CW-230 belonging to Zn 2p_3/2_ and Zn 2p_1/2_ (Appendix A). Compared with Zn@CW, after oxidation, both peaks have a slight shift. As shown in Figure 3d, the high-resolution spectrum of C 1s has four peaks at 284.3, 284.9, 287.5, and 289.0 eV, corresponding to the characteristic signals of C-C, C-O/C-N, C=O, and O-C=O bonds, respectively [38,39]. It can be seen from Appendix A that the oxidation process in the air greatly increases the oxygen-containing functional groups on carbonized wood, which may improve the hydrophilicity of electrode materials. The high-resolution Co spectrum is fitted by Co^0^, Co^2+^, Co^3+^, and satellite peaks, as shown in Figure 3e and Appendix A. The peak at 780.6 eV in the Co 2p_3/2_ region belongs to oxidized Co species (Co^2+^). The main peak of Zn-Co@CW at 778.15 eV belongs to Co 2p_3/2_, which comes from the zero-valence state of Co. The main peak of Zn-Co_3_O_4_@CW-230 at 779.6 eV belongs to Co 2p_3/2_, which comes from Co^3+^ [36,40,41]. This indicates that the simple substance Co is completely oxidized to Co_3_O_4_ with a better energy storage effect. 

The N_2_ adsorption and desorption tests reveal the coexistence of many micropores/mesopores in the Zn-Co_3_O_4_@CW-230 (Figure 3f). These pores are critical for ion adsorption and diffusion in electrochemical processes [42]. As shown in Appendix A, the adsorption isotherm of Zn-Co_3_O_4_@CW-230 is type IV, and the hysteresis loop of adsorption isotherm is type H4, which has a clear micropore filling phenomenon at low pressure [35]. The generation of type H4 hysteresis loop shows that mesopores increase, which is consistent with the data of the aperture distribution diagram. The specific surface area of Zn-Co@CW and Zn-Co_3_O_4_@CW-230 are 270.6 and 300 m^2^ g^−1^, and the pore size distribution is mainly located at 1.0, 2.8, and 3.8 nm, respectively. These prove that Zn-Co_3_O_4_@CW-230 has more micropores and mesopores, a larger specific surface area, and may obtain greater energy storage capacity.

All sample electrodes were subjected to cyclic voltammetry (CV), and galvanostatic charge-discharge (GCD) tests on the three-electrode system (Figure 4, Appendix A). Cyclic voltammetry tests are performed on the electrodes at scan rates of 5, 10, 15, 20, and 30 mV s^−1^, and the CV curve of all samples demonstrates a slightly distorted quasi-rectangular shape, indicating the formation of electrical double layers with representative electrostatic ion sorption and desorption [43]. The current response value and loop area of the Zn-Co_3_O_4_@CW-230 in the voltage window of −1~0 V are the largest, indicating that the Zn-Co_3_O_4_@CW-230 has the highest capacity. GCD testing of all sample electrodes at current densities of 5~30 mA cm^−2^, the ideal isosceles triangle charge-discharge curve reveals the good electrochemical performance and fast mass transfer rate of wood-derived carbon materials. The area-specific capacitances of Zn-Co_3_O_4_@CW-230 are 7.83, 7.47, 7.05, 6.82, and 6.46 F cm^−2^ at current densities of 5, 10, 15, 20, and 30 mA cm^−2^, respectively (Figure 4b). The area-specific capacitances of CW-230, Zn@CW-230, and Co_3_O_4_@CW-230 are 2.32, 6.03, and 4.46 F cm^−2^ at 10 mA cm^−2^, respectively (Appendix A). This is due to the high conductivity of wood-derived carbon substrate and easy electrolyte penetration, and better utilization of Co_3_O_4_ electroactive surface in the redox reaction [44,45], which is consistent with the results inferred from the previous characterization. In Figure 4d, the electrochemical performances were further explored by EIS, and the Nyquist plots were fitted with an equivalent circuit diagram (Appendix A). R_s_ is the internal solution resistance, including the inherent resistance of the material, the ionic resistance of the electrolyte, and the contact resistance between the material and the electrolyte. R_ct_ represents the charge transfer resistance [46], and W is the Wobbelli impedance represented by the line slope in the low-frequency region, which affects the diffusion rate of electrolyte ions in the electrode material. The Nyquist plots of Co@CW, Co_3_O_4_@CW-230, Zn-Co@CW, and Zn-Co_3_O_4_@CW-230 with R_Ω_ resistance values of 3.00, 2.61, 3.35, and 2.98 Ω, respectively, show extraordinary conductivity.

To investigate the application potential of CW-230, Co_3_O_4_@CW-230, and Zn-Co_3_O_4_@CW-230 electrodes in liquid-state SCs, we assembled CW-230//CW-230, Co_3_O_4_@CW-230//Co_3_O_4_@CW-230, and Zn-Co_3_O_4_@CW-230//Zn-Co_3_O_4_@CW-230 symmetrical SCs. Figure 5a,b and Appendix A are the GCD test and CV test of the above SCs. It can be seen from the test results that the CV curves of CW-230//CW-230 and Co_3_O_4_@CW-230//Co_3_O_4_@CW-230 are not regular rectangles, indicating that their electric double layer energy storage behavior has changed. Moreover, the areal-specific capacities of the CW-230//CW-230 and Co_3_O_4_@CW-230//Co_3_O_4_@CW-230 SCs are only 0.618 F cm^−2^ at 2.0 mA cm^−2^ and 2.35 F cm^−2^ at 5.0 mA cm^−2^. However, the CV curves of Zn-Co_3_O_4_@CW-230//Zn-Co_3_O_4_@CW-230 SC at different scan speeds (Figure 5a) still maintain an ideal rectangle, which is due to the surface of Zn-Co_3_O_4_@CW-230 electrode has higher conductivity and mechanical stability, benefiting from the semi-embedded design and more stable structure of Zn-Co_3_O_4_/C-230 particles. The area-specific capacitance of Zn-Co_3_O_4_@CW-230//Zn-Co_3_O_4_@CW-230 SC reaches 2.61, 2.39, 2.24, 2.11, and 1.89 F cm^−2^ at 5, 10, 15, 20 and 30 mA cm^−2^, respectively (Figure 5c), maintaining excellent rate performance. This SC also exhibits the superior energy densities of 0.36 mWh cm^−2^ (6.0 mWh cm^−3^) at the power densities of 2.5 mW cm^−2^, which surpasses CW-230//CW-230 and Co_3_O_4_@CW-230//Co_3_O_4_@CW-230 (Figure 6). It is worth noting that the 0.6 mm thick monolithic electrode is also an important performance indicator when assembling the SC. In order to further evaluate the energy storage performance of materials, the data of energy/power density of different SC devices are compared in Appendix A [39,47,48,49,50,51,52,53]. The excellent energy density of Zn-Co_3_O_4_@CW-230//Zn-Co_3_O_4_@CW-230 SC is significantly higher than other SC equipment. Furthermore (Figure 5d), the capacitance of Zn-Co_3_O_4_@CW-230//Zn-Co_3_O_4_@CW-230 SC remains at 97.3% after cycling for 10,000 cycles at 50 mA cm^−2^. The capacities of CW-230//CW-230, Co_3_O_4_@CW-230//Co_3_O_4_@CW-230, and Zn@CW-230//Zn@CW-230 SCs fluctuated greatly during long cycles at high current densities, and their capacity retention ratios fluctuated in the range of 90~105%. The results show that the symmetric SC assembled with Zn-Co_3_O_4_@CW-230 electrode exhibits excellent electrical conductivity, ion transport rate, and cycling stability.

## 3. Materials and Methods

### 3.1. Materials

Cobalt nitrate hexahydrate (Co(NO_3_)_2_∙6H_2_O, 99%), Zinc nitrate hexahydrate (Zn(NO_3_)_2_∙6H_2_O, 99%), 2-methylimidazole (2-MI), sodium hydroxide (NaOH), and Potassium hydroxide (KOH) were obtained from Aladdin and used directly. The Poplar wood block used in this study was received from a wood processing factory in the south of China. All reagents are of analytical-grade and used without further purification. Deionized (DI) water was used as the solvent.

### 3.2. Wood Pretreatment

Poplar monoliths were cut into cuboids with the dimensions 40 × 40 × 4 mm^3^ (radial × tangential × longitudinal, *R* × *T* × *L*). The slices were then immersed in a 15 wt% NaOH aqueous solution for 1 h. Afterward, the slices were thoroughly rinsed with deionized water until the pH reached 9. Finally, the pretreated wood was dehydrated by freeze-drying.

### 3.3. Preparation of HZ@Wood

The pretreated wood slices were immersed in a mixed solution for 2 h, which was prepared by dissolving Co(NO_3_)_2_∙6H_2_O (1.16 g) and Zn(NO_3_)_2_∙6H_2_O (0.59 g) in methanol/deionized water (30/5 mL). A 2-MeIm solution prepared by dissolving 9.85 g of 2-MI in methanol/deionized water (30/5 mL) was dropwise added and stirred at ambient temperature for 24 h. The HZ/Wood was washed with methanol and freeze-dried in a vacuum. ZIF-8@Wood and ZIF-67@Wood were developed in the same way.

### 3.4. Preparation of Zn-Co@CW and Zn-Co_3_O_4_@CW-230

HZ@Wood was put into a tube furnace under temperature programming and constant Ar flow. The sample was heated slowly from room temperature to 700 °C with a heating rate of 5 °C/min, maintained for 5 h under an Ar atmosphere. After that, the furnace was naturally cooled to room temperature in an Ar atmosphere to obtain black powders, denoted as Zn-Co@CW. Then, Zn-Co_3_O_4_@CW-230 was obtained by further calcining Zn-Co@CW in an air atmosphere at 230 °C for 1 h at the same heating rate. In addition, we prepared Co@CW, Zn@CW, CW, Co_3_O_4_@CW-230, Zn@CW-230, and CW-230 from ZIF-67@Wood and ZIF-8@Wood and pure wood by the same preparation method.

### 3.5. Characterization and Electrochemical Testing

The morphology and structure were characterized by scanning electron microscopy (SEM, Hitachi SU8010). The phase composition of as-obtained materials was analyzed by an X-ray diffraction instrument (XRD, Bruker D8). The surface chemical information was detected by X-ray Photoelectron Spectroscopy (XPS, ThermoFisher ESCALAB 250Xi). The specific surface area and hierarchical porous structure were measured by automatic surface area and porosity analysis (Brunauer–Emmett–Teller (BET), MicroActive ASAP2020). Raman spectroscopy (Raman, LabRAM HR Evolution) was used to test the chemical states of the carbon atoms. Thermogravimetric (TG) analyses were carried out on a thermogravimeter (TGA5500) from room temperature to 800 °C in the air with a heating rate of 10 °C min^−1^.

The electrochemical performances of a single electrode were tested with a three-electrode setup on an electrochemical workstation (Vertex.One/Vertex.C, IVIUM, Eindhoven, The Netherlands) by using an Hg/HgO as the reference electrode, a platinum plate as the counter electrode, and a 6M KOH aqueous solution as the electrolyte. As shown in Appendix A, 6M KOH aqueous solution is also used as an electrolyte to assemble a two-electrode device to test the liquid-state symmetric SCs.

## 4. Conclusions

In conclusion, we found that HZs could be used as precursors to obtain a monolithic electrode with semi-embedded active substances on the surface. The inner and outer surface structure of wood-derived carbon was modified by derived Zn-Co_3_O_4_/C-230 particles, which greatly improved the specific surface area of carbonized wood and the electrochemical and mechanical stability of electrodes. In addition, the step of oxidation in the air is also crucial. It not only improves the crystallinity of carbon-reducing electrode resistance but also will obtain TMOs with better energy storage effect. Based on the above advantages, the area-specific capacitance of Zn-Co_3_O_4_@CW-230 can reach 7.83 F cm^−2^ at current densities of 5.0 mA cm^−2^ (0.6 mm thick). The areal-specific capacitance of assembled Zn-Co_3_O_4_@CW-230//Zn-Co_3_O_4_@CW-230 SC is as high as 2.61 F cm^−2^ at 5.0 mA cm^−2^, and it shows great energy densities of 0.36 mWh cm^−2^ (6.0 mWh cm^−3^), due to its high-capacity contribution of Co_3_O_4_, unique semi-embedded structure, and conductivity. The strategy of semi-embedding the TMOs derived from ZIFs into wood-derived carbon is expected to promote the development of other ZIFs and wood composites and provide a new research idea for the large-scale production of high-performance charcoal self-supporting composite electrodes and the practical application of high energy density SCs.

## Figures and Tables

**Figure 1 molecules-27-08572-f001:**
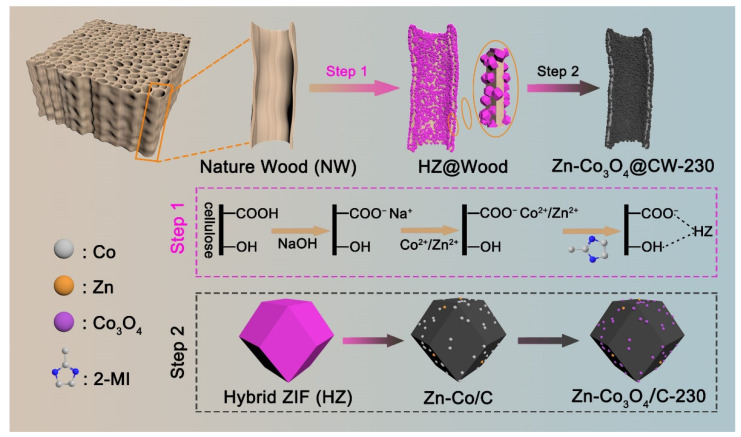
Schematic representation of the fabrication procedure for Zn-Co_3_O_4_@CW-230.

**Figure 2 molecules-27-08572-f002:**
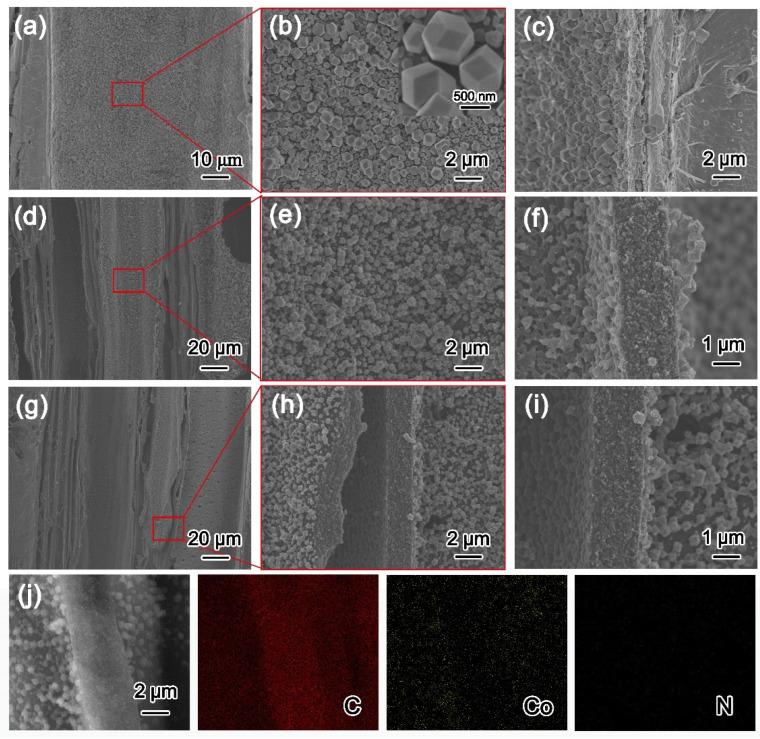
SEM images of HZ@Wood (**a**–**c**), Zn-Co@CW (**d**–**f**), and Zn-Co_3_O_4_@CW-230 (**g**–**i**); C, Co, Zn mapping of Zn-Co_3_O_4_@CW-230 (**j**).

**Figure 3 molecules-27-08572-f003:**
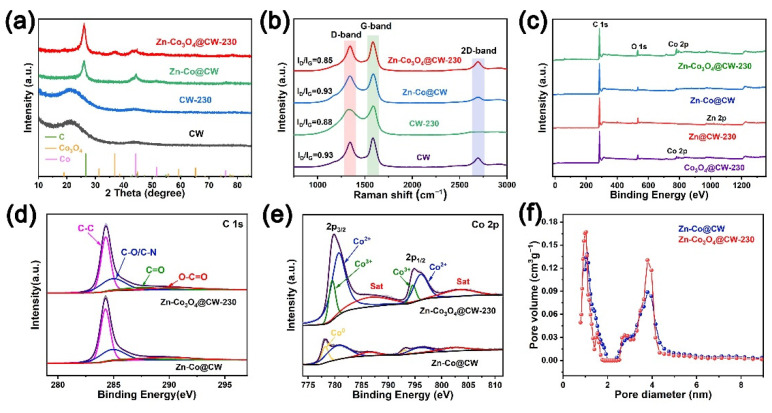
(**a**) XRD spectra of CW, CW-230, Zn-Co@CW, and Zn-Co_3_O_4_@CW-230; and Raman spectra (**b**). XPS spectra of (**c**) full scan survey for Co_3_O_4_@CW-230, Zn@CW-230, Zn-Co@CW, and Zn-Co_3_O_4_@CW-230. High-resolution XPS spectra of Zn-Co@CW and Zn-Co_3_O_4_@CW-230 for (**d**) C 1s and (**e**) Co 2p; and (**f**) corresponding pore size distribution.

**Figure 4 molecules-27-08572-f004:**
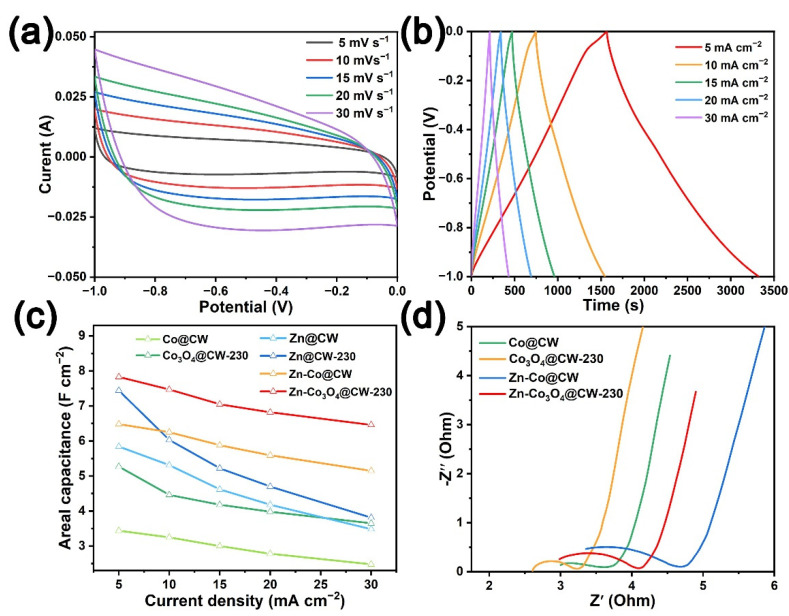
Cyclic voltammograms (CVs) at different scan rates and galvanostatic charge-discharge (GCD) curves at different current densities of (**a**,**b**) Zn-Co_3_O_4_@CW-230. (**c**) Specific capacitances of Zn/Co/Zn-Co@CW and Zn/Co_3_O_4_/Zn-Co_3_O_4_@CW-230 at different current densities. (**d**) the EIS plots of the Co/Zn-Co@CW and Co_3_O_4_/Zn-Co_3_O_4_@CW-230.

**Figure 5 molecules-27-08572-f005:**
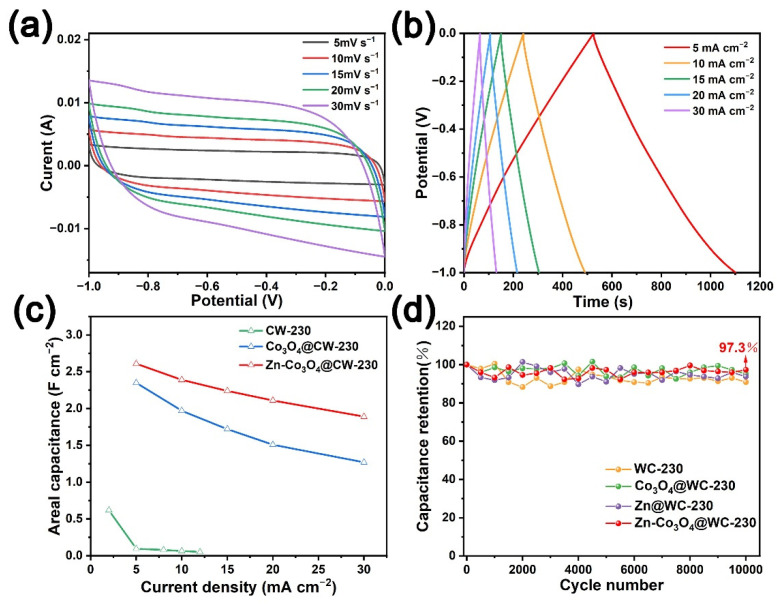
Capacitive performance of the SC of Zn-Co_3_O_4_@CW-230: (**a**) cyclic voltammograms at various scan rates in a two-electrode system, and (**b**) GCD profiles at different current densities. (**c**) Rate performances based on the GCD curves at different current densities, (**d**) the SC device measured at 50 mA cm^−2^ for 10,000 cycles.

**Figure 6 molecules-27-08572-f006:**
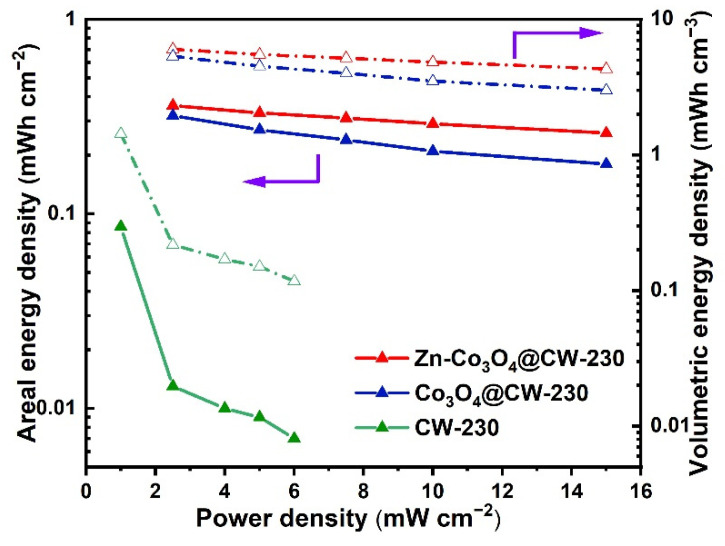
Areal energy density and volumetric energy density at different power densities of the SC of CW-230, Co_3_O_4_@CW-230, and Zn-Co_3_O_4_@CW-230.

## Data Availability

The data used to support these findings have been included in this article. Additional information is available from the corresponding authors upon request.

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
