# Peer review of "Semi-Embedding Zn-Co3O4 Derived from Hybrid ZIFs into Wood-Derived Carbon for High-Performance Supercapacitors"

_molecules, 2022, doi:10.3390/molecules27238572_

Round 1
Reviewer 1 Report
The manuscript entitled "Semi-Embedding Zn-Co3O4 Derived from Hybrid ZIFs into Wood-derived Carbon for High-performance Supercapacitors" reported hybrid ZIFs derived Zn doped Co3O4/carbon (Zn-Co3O4/C) particles semi-embedded in wood-derived carbon skeleton for integrated electrodes. The prepared Zn-Co3O4@CW-230 electrode displays ultrahigh area specific capacitances of 7.83 F cm-2 at the current densities of 5 mA cm-2. In general, this work is interesting and worthy of being featured, but there are still some issues that need to be pointed out.
1. Some sentences in the manuscript are lengthy and unclear. Paying particular attention to English grammar, spelling, and sentence structure, so that the goals and results of the study are clear to the reader.
2. It is interesting to incorporate ZIF or MOF into wood. Some relevant, important and recent articles should be included to support this article: MOFs meet wood: reusable magnetic hydrophilic composites toward efficient water treatment with super-high dye adsorption capacity at high dye concentration; When MOFs meet wood: From opportunities toward applications; ZIF-67/wood derived self-supported carbon composites for electromagnetic interference shielding, sound and heat insulations.
3. When providing the structure of wood, more relevant articles should be added.
4. The sentences of reasoning in the section of results and discussion should add references to enhance credibility.
5. Changing the color of the ruler in Figure 2 to white will make it clearer.
6. The capacitive contribution of Zn-Co3O4@CW-230 at different scan rates should be analyzed in the part of evaluating the energy storage mechanism.
7. A table or a figure should be added at the end of the results and discussion to compare the capacitive performance of Zn-Co3O4@CW-230 with other wood derived carbon or composite carbon electrodes for supercapacitors, such as: Polymers 14 (13), 2521, 2022; Biochar, 2022, 4(1): 1-19; Wood Science and Technology 56, 1191–1203, 2022; Small 18 (25), 2201307, 2022; Small 17 (35), 2102532, 2021; Advanced Functional Materials 31 (31), 2101077, 2021; Chemical Engineering Journal 414 (2021): 128767; etc.
8. To have a better readability, the experimental section is suggested to shift ahead of results and discussion part.
9. There are many problems with the format of references, such as the case of the first letter of each word in the title, the superscript and subscript format in compounds, and the abbreviation of journal names. Please standardize the format carefully.
Reviewer 2 Report
Reviewer’s comments
Manuscript Number: molecules-2043902
Title: Semi-Embedding Zn-Co3O4 Derived from Hybrid ZIFs into Wood-derived Carbon for High-performance Supercapacitors
Journal: molecules
- Since the prepared materials are nanocomposites, the elemental composition (Zn:Co:C) should be provided using EDX of ICP data to make sure the experimental ratios are agreed with the nominal ratios.
- The findings (Capacitance, energy density, etc.) should be compared with other related materials. A detailed table of comparison should be provided.
3. EIS data (Fig. 4d) should be fitted with an equivalent circuit, and the electrochemical parameters should be obtained and discussed.
4. The CV and CDC (Fig. 4 a, b) should be compared with the materials counterparts (Zn@WC, Co@WC, ….) same as Fig. 4c.
5. High-resolution XPS spectra of Zn should be provided and discussed.
6. For symmetric supercapacitor (Fig. 5 a,b), it is recommended to avoid the negative potential window. The practical supercapacitor (Full cell) should be tested in positive potential. Otherwise, it is like reversing the electrode polarity which can be dangerous and detrimental to the safety of the device.
7. It is recommended to show the stability every 50 or 100 cycles, 500 cycles are a wide range (Fig. 5d).
- The correlation between structural/morphological findings and electrochemical performance should be discussed and compared with other related composites such as: Journal of Solid State Electrochemistry, 18 (9) (2014) 2505–2512, Journal of Materials Science, 54(1) (2019) 683–692
Round 2
Reviewer 1 Report
Accept in present form
Reviewer 2 Report
The authors have addressed most of the comments in the revised version. The current version could be accepted for publication.